# The Sentinel Lymph Node in Treatment Planning: A Narrative Review of Lymph-Flow-Guided Radiotherapy

**DOI:** 10.3390/cancers15102736

**Published:** 2023-05-12

**Authors:** Loic Ah-Thiane, Caroline Rousseau, Maud Aumont, Axel Cailleteau, Mélanie Doré, Augustin Mervoyer, Loig Vaugier, Stéphane Supiot

**Affiliations:** 1Department of Radiotherapy, ICO René Gauducheau, Boulevard Jacques Monod, 44800 St-Herblain, France; maud.aumont@ico.unicancer.fr (M.A.); axel.cailleteau@ico.unicancer.fr (A.C.); melanie.dore@ico.unicancer.fr (M.D.); augustin.mervoyer@ico.unicancer.fr (A.M.); loig.vaugier@ico.unicancer.fr (L.V.); stephane.supiot@ico.unicancer.fr (S.S.); 2Department of Nuclear Medicine, ICO René Gauducheau, Boulevard Jacques Monod, 44800 St-Herblain, France; caroline.rousseau@ico.unicancer.fr; 3CRCI2NA, UMR 1307 Inserm—UMR 6075 CNRS, Nantes University, 44000 Nantes, France; 4CRCI2NA, Inserm UMR 1232, CNRS ERL 6001, Nantes University, 44000 Nantes, France

**Keywords:** sentinel lymph node, lymphatic drainage, radiation therapy, treatment planning, lymph-flow-guided radiotherapy, precision medicine

## Abstract

**Simple Summary:**

The sentinel lymph node is a surgical technique developed in oncological surgery to identify and analyze fewer lymph nodes than a conventional lymph node dissection in order to limit the morbidity and mortality of such an extensive procedure without compromising the patients’ outcomes. This concept seems to also be useful in radiation oncology that treats lymph node areas. This may help radiation oncologists to treat their patients more precisely by targeting more accurately pathological sites and sparing healthy tissues. The aim of this review is to highlight the feasibility and level of proof regarding the use of this technique for treatment planning in radiation oncology.

**Abstract:**

The sentinel lymph node technique is minimally invasive and used routinely by surgeons, reducing the need for morbid extensive lymph node dissections, which is a significant advantage for cancer staging and treatment decisions. The sentinel lymph node could also help radiation oncologists to identify tumor drainage for each of their patients, leading to a more personalized radiotherapy, instead of a probabilistic irradiation based on delineation atlases. The aim is both to avoid recurrence in unexpected areas and to limit the volume of irradiated healthy tissues. The aim of our study is to evaluate the impact of sentinel lymph node mapping for radiation oncologists. This concept, relying on sentinel lymph node mapping for treatment planning, is known as lymph-flow-guided radiotherapy. We present an up-to-date narrative literature review showing the potential applications of the sentinel lymph node technique for radiotherapy, as well as the limits that need to be addressed before its routine usage.

## 1. Introduction

The lymphatic vasculature is a system draining the organs through vessels and lymph nodes that act as filters. Its complexity stems from the numerous anatomical variants described in studies during cadaver dissections or surgical explorations, but knowledge of these variations is becoming increasingly available [1]. The metastatic spreading of cancer was described in some models to start with the invasion of the first draining loco-regional nodes, which then became known as sentinel lymph nodes (SLNs) [2]. The idea of using these SLNs thus emerged, based on a sentinel lymph node mapping (SLNM) using colorimetric, fluorescent or radioisotope tracers to identify the area of drainage of an anatomical territory, followed by sentinel lymph node biopsies (SLNBs) to assess whether a node is pathological (pSLN) or not (nSLN) [3]. The concept of SLNs was a major breakthrough for surgeons, allowing for a decrease in the number of morbid extensive lymph node dissections (ELND). SLNB is a minimally invasive technique, used routinely in penile cancer, breast cancer and melanoma thanks to its reliable performance and proven safety, and plays a key role in cancer staging and treatment decisions [4,5].

In order to better personalize radiotherapy treatments, we hypothesized that SLNM and SLNB could be used to provide more personalized radiotherapy by identifying the drainage of each patient’s tumor, making it possible to not miss unexpected areas or to preserve healthy tissues. This concept is called “lymph-flow-guided radiotherapy” (LFGRT), and we aim to present a current review of the rather sparse literature exploring this hypothesis. We will discuss localizations of cancers for which the SLN has been evaluated in surgery and in which radiotherapy could play a role, both in well-established areas such as head and neck cancers, or areas currently under investigation such as renal cancers.

## 2. Breast Cancer

### 2.1. Axillary Lymph Node Dissection (ALND) Can Be Avoided Thanks to SLNB

In patients with nSLN, ALND can be avoided, as no difference was observed in overall and disease-free survival, or in axillary failure, which was low and reported in 0.7–0.8% of patients [6,7].

In patients with pSLN presenting a tumor smaller than 5 cm and no palpable adenopathy (cT1-T2 cN0), ALND could be avoided in patients with micrometastasis (<2 mm), since the IBCSG 23-01 trial showed no inferiority in disease-free survival after 10 years [8], as well as in patients with up to two macrometastasis but no capsular effraction, since the ACOSOG Z0011 (Alliance) trial showed no inferiority in overall survival after 10 years [9]. ALND was also proposed to be replaced by axillary radiotherapy, since the AMAROS and OTOASOR trials showed no difference in overall and disease-free survival between ALND and axillary radiotherapy [10,11]. A retrospective study compared 260 patients who received axillary radiotherapy versus those who did not and found no significant difference: 5-year overall survival was 93.4% versus 96.8% (*p* = 0.19), respectively, and 5-year disease-free survival was 92.3% versus 100% (*p* = 1.06), respectively [12].

A systematic review highlighted that ALND induced significantly more lymphedemas and shoulder dysfunctions in comparison with observation or axillary radiotherapy [13]. For most patients with nSLN or with pSLN (up to two metastasis) and cT1-T2 cN0 tumors, ALND should be avoided to decrease morbidity. Axillary radiotherapy is worth discussing in case of risk factors.

### 2.2. SLNM/SLNB Indicates Nodal Irradiation

Regional nodal irradiation, in addition to breast/chest wall irradiation, is currently indicated in case of clinical or pathological node involvement but deserves to be challenged. In fact, two phase III trials demonstrated that nodal irradiation (including axillary, infra/supraclavicular, internal mammary nodes) reduced breast cancer recurrence and specific mortality but did not significantly improve overall survival [14,15]. Moreover, a recent trial randomized 735 patients who received nodal irradiation, both including and excluding the internal mammary nodes, and found no benefit to irradiating this area, with the exception of a subgroup of patients with medial/central tumors [16], explained by their pattern of lymphatic drainage. Some authors suggested performing SLNM with the acquisition of SPECT/CT images to identify the drainage of each tumor, knowing that up to 50% of patients can present drainage in both axillary and internal mammary nodes, depending on the tumor’s location [17]. Figure 1 shows two examples of drainage in internal mammary nodes. However, in practice, only axillary nodes are noticed because they matter in surgery. Given recent findings, knowing the specific drainage of the cancer would help radiation oncologists delineate lymphatic areas, notably the internal mammary nodes, for relevant prophylactic irradiation [18].

While SLNM identifies lymphatic drainage of the breast tumor after peritumoral tracer injection, axillary reverse mapping (ARM) identifies drainage of the upper limb after arm injection. ARM was initially developed for surgeons to preserve the main nodes draining the arm and not the tumor during ALND 6, but ARM seemed to be applicable for axillary irradiation too [19]. A dosimetric evaluation pointed out that all the nodes identified by ARM received the prescribed dose during standard axillary radiotherapy, explaining the rate of arm lymphedema [20]. A pilot study showed the feasibility of combining SLNM and ARM to preserve the main nodes, draining the arm while conserving the good coverage of the SLN sites in 5/6 of the patients. In the remaining patient, it was not possible to preserve these nodes because the SLNM and ARM overlapped [21]. The next step is to conduct trials to evaluate the oncological outcomes and their impact on lymphedema when reducing axillary irradiation volumes.

### 2.3. The Role of SLNB Needs to Be Redefined in a Neoadjuvant Setting

In cN0 patients, SLNB after neoadjuvant chemotherapy demonstrated a comparable performance to SLNB in upfront surgery and reduced the need to perform an ALND [22].

In patients with nodes confirmed by histology, the SN FNAC trial validated SLNB after neoadjuvant chemotherapy [23]. In case of residual nodal disease (ypN+), the guidelines recommend treating the axillary nodes [24]. For these patients, the ongoing ALLIANCE A011202 trial aims to determine the optimal treatment by comparing ALND and axillary radiotherapy with axillary radiotherapy alone (ClinicalTrials.gov number: NCT01901094). In case of complete nodal response (ypNO), the need for adjuvant nodal treatment is more debatable; hence, the ongoing NSAPB B-51 trial compares nodal irradiation with observation (ClinicalTrials.gov number: NCT01872975).

In conclusion, there is a clear decrease in ALND in breast cancer thanks to SLNB, in cases of nSLN but also in selected cases of pSLN. The results of SLNB indicate nodal irradiation, but SLNM may also provide some information on specific tumor drainage (especially internal mammary drainage) to help define which volumes should be targeted in radiotherapy. Moreover, ARM identifies the lymphatic nodes that drain the arm instead of the tumor and is worth exploring to reduce radiation-induced lymphedema. LFGRT is thus appealing as an effective method of irradiation with lower toxicity.

## 3. Gynecologic Cancers

### 3.1. SLNB Is a Well-Documented Technique in Vulvar Cancers

Locally advanced vulvar carcinomas are usually treated conservatively thanks to chemoradiation, whereas early-stage treatment consists of radical resection with nodal assessment and can be followed by adjuvant radiotherapy. Lymph node staging is a major prognostic factor in vulvar cancers [25]. For FIGO IB to II and lateral lesions (≥2 cm from vulvar midline) with clinically/radiologically node negative tumors, SLNB is recommended, since nSLN is associated with low morbidity, groin recurrence and disease-specific mortality, while being more cost-effective than extensive lymphadenectomy [26].

In case of pSLN, the management of ipsilateral groin with lymphadenectomy and radiotherapy should be discussed [27]. The GROINSS-V-II trial studied 322 patients with pSLN to evaluate whether groin dissection could be replaced by inguinofemoral radiotherapy. Due to high groin recurrence, the protocol had to be amended to allow for patients with SLN > 2 mm (macrometastasis) to undergo lymphadenectomy as the standard of care, but patients with SLN ≤ 2 mm (micrometastasis) could continue to receive radiotherapy. The 2-year groin recurrence rate was low for patients with micrometastasis (1.6%), but high for patients with macrometastasis when treated by radiotherapy (22%) compared to those treated by lymphadenectomy (6.9%). Ipsilateral inguinofemoral irradiation appears to be a low-morbidity option for patients with micrometastasis but should not be the first intention in case of macrometastasis [28].

How to manage contralateral groin remains unclear. Two retrospective monocentric studies suggested not treating contralateral groin, since they found very low rates of contralateral involvement: 0% (0/28) patients and 5.3% (1/19) patients, respectively [29,30]. However, a recent study reported a higher rate of contralateral involvement, at 22.2% (4/18) of patients, after an initial diagnosis of unilateral metastasis, supporting current guidelines in favor of contralateral prophylactic treatment by either lymphadenectomy or radiotherapy [31].

For larger tumors (greater than 4 cm), the negative predictive value deteriorates, so there is no strong evidence to recommend using the SLN technique [30].

### 3.2. SLNB Is Not the Standard Reference for Node Staging in Cervical Cancers at Present

Lymph node status leads the indication for radiotherapy in cervical cancers. The treatment is exclusively chemoradiation if metastatic lymph nodes are detected before radical surgery, or adjuvant chemoradiation if detected after resection. SLNB is currently employed in addition to pelvic node dissection but not alone, despite some interesting performances [32]. Indeed, questions have been raised about the ability to detect micrometastasis, reliability in intraoperative detection and the limited evidence obtained from prospective studies [33]. The SENTIX trial evaluated intraoperative SLN frozen section and SLNB without pelvic node dissection in 395 patients: SLN pathological examinations achieved high detection for node staging, but the intraoperative SLN frozen section failed to detect about 50% of pathological nodes [34]. Ongoing SENTICOL III and PHENIX trials are enrolling patients with early-stage cervical cancer. The SENTICOL III trial follows the SENTICOL II trial, which showed the decreased morbidity of SLNB alone [35] and randomizes patients between SLNB alone (experimental arm) and SLNB plus pelvic node dissection (reference arm). In the PHENIX trial, all patients undergo SLNB and are allocated into either the PHENIX-I (if nSLN) or PHENIX-II (if pSLN) cohorts. Patients in each cohort are randomized after the SLNB between observation (experimental arm) and pelvic node dissection (reference arm). The primary outcome of these two trials is disease-free survival to demonstrate non-inferiority, and results are expected in 2026 [36,37].

For more advanced cervical cancers, higher than FIGO 2018 stage Ib3, the involvement of para-aortic nodes needs to be assessed to guide irradiation volumes. This assessment is based on FDG PET-CT and para-aortic lymphadenectomy. The role played by SLNB is little documented and thus cannot be recommended [38,39].

### 3.3. SLNB Could Guide the Indication of Adjuvant Radiotherapy in Endometrial Cancers

For endometrial cancers, decisions regarding adjuvant treatments such as radiotherapy, brachytherapy or chemotherapy is influenced by nodal assessment, but also depend on other histopathological factors and, more recently, on molecular and genomic profiles [40,41]. For nodal assessment, a Cochrane meta-analysis validates SLNB as an accurate technique with high sensitivity [42], and some researchers have developed an algorithm using data from 247 patients, including SLNB, that can identify the involved lymph nodes with a sensitivity of 98% and negative predictive value of 99.5% to replace extensive lymphadenectomy [43]. A recent meta-analysis pooled the results from 14 studies evaluating SLNB and analyzing over 2000 patients with low- and intermediate-risk endometrial cancers and found about 10% of pathological involvement with a high detection rate and negative predictive value [44]. In high-risk cancers, the use of SLNB is more discussed [45]. Nevertheless, a multi-institutional retrospective study showed similar disease-free survival and overall survival for patients undergoing SLNB with and without back-up lymphadenectomy [46], and a review identified only retrospective studies, but suggested the non-inferiority of SLNB compared to lymphadenectomy [47]. Thanks to these good performances, SLNB could guide the indication of adjuvant radiotherapy. For example, a study showed that the decision of adjuvant radiotherapy changed for some patients based on the SLNB results [48].

In conclusion, lymph node staging is a major prognosis factor for vulvar cancers. For FIGO IB to II, <4 cm, and cN0 tumors, SLNB is recommended. In case of pSLN, ipsilateral groin should be treated by either radiotherapy (if micrometastasic disease) or lymphadenectomy (if macrometastatic disease), whereas the treatment of contralateral groin is more open to discussion. In cervical cancers, SLNB alone cannot be recommended in routine treatment at present, and complements the pelvic node dissection that remains the standard until the results of ongoing trials are reported. In endometrial cancers, SLNB could help in the radiotherapy decision-making process.

## 4. Urologic Cancers

### 4.1. Penile Cancers Represent a Leading Indication of SLNB

In penile cancers, SLNB is a highly recommended procedure for the management of clinically node-negative patients based on the European Association of Urology guidelines [49]. Systematic reviews have confirmed the relevance of SLNB in this cancer, which has a very stereotyped echelon-based pattern of lymphatic drainage [50,51]. SLNs are detected during surgery with a high sensitivity and specificity of about 77% and 100%, respectively [52], especially when using blue dye and radiotracer in combination [53]. Performances appear even better when acquiring 3D-imaging in SPECT/CT before surgical detection, to increase the detection rate [54] and decrease the rate of false-positive nodes [55].

However, no studies investigated the use of SLNM and SLNB in radiation oncology, mainly because the benefits of nodal irradiation have not been demonstrated. In the absence of nodal involvement, prophylactic inguinal irradiation at 50 Gy showed no decrease in recurrences compared to surveillance [56]. If lymph nodes are involved, inguinal dissection is performed, and adjuvant radiotherapy might be offered in case of bad prognosis factors. The use of adjuvant radiotherapy is under debate since a systematic review showed no benefits, and thus a standard recommendation cannot be made [57].

### 4.2. SLNB Is Non-Mature in Bladder, Testicular, and Renal Cancers

SLNB has been described in bladder cancers for more than 20 years, but still presents non-negligible rates of false-negative lymph nodes, thus requiring further investigation, notably regarding radiotracers and detection techniques [58,59].

In testicular cancers, SLNB appears safe in prospective studies, but its value for guiding adjuvant treatment remains to be demonstrated [60,61].

In renal cancers, SLNM is not an easy technique to reproduce because it can be non-contributory in 30% of cases due to a lack of drainage of the radiotracer through lymphatic vessels [62]. Aside from these technical difficulties, its ability to detect and then treat lymph nodes is debated since it does not seem to change overall survival according to a recent meta-analysis [63].

### 4.3. SLNB Could Help Redefine Irradiation Volumes in Prostate Cancers

In prostate cancers, extended pelvic lymph node dissection remains the gold standard for lymph node staging, but SLNB demonstrated comparable results with high sensitivity and specificity and a low rate of false-negative nodes [64]. Nodal staging was improved, with up to 94% accuracy achieved with the combination of SLNB and a PSMA PET-CT [65]. Although not used routinely [66], the SLNB appears to be an accurate technique with low morbidity that could help in treatment decisions in intermediate- and high-risk prostate cancers [67]; for instance, to extend androgen deprivation therapy in cases where pathological nodes were detected or to better define volumes in radiotherapy [68].

Prophylactic pelvic elective node irradiation (ENI), in addition to prostate gland irradiation, has indications for unfavorable intermediate- and high-risk cN0 prostate cancers [69], even if the benefits of pelvic irradiation remain controversial [70,71]. Some research organizations, such as the RTOG or the UK CRUK PIVOTAL, have suggested delineation atlases to contour lymph node areas that classically include the distal common iliac, external and internal iliac, and obturator lymph nodes [72]. The French group GETUG identified some rarely irradiated regions, which are nevertheless at risk of invasion [73], such as the proximal common iliac, para-rectal, peri-vesical, peri-vesicular, pre-sacral, pudendal, inguinal and retroperitoneal drainage regions, as described in SPECT-CT [74], and more recently in PET-CT [75,76,77]. The undercoverage of these regions could explain some patterns of disease recurrence in proximal common iliac [78] or retroperitoneal and inguinal regions [79], whereas in-field recurrence is observed less often [80]. The difficulty of obtaining an overall recommendation could come from the fact that lymphatic drainage varies considerably depending on the intraprostatic localization of the tumor (base or apex, ventral or dorsal, central or lateral) [81].

A phase I proof-of-concept study evaluated SLNM’s ability to include relevant lymph nodes in the target volume in six patients for a more personalized irradiation [82]. A phase II study showed good biochemical control (73.8%) at 5 years for 61 patients when the mapped drainage areas were irradiated in addition to the usually recommended areas [83]. An in silico study simulated radiotherapy plans according to RTOG delineation guidelines in 57 patients who had an SLNM procedure from a previous trial: 305 pelvic nodes were identified (mean of 5.4 “hot” nodes per patient); 67/305 (22%) would not have been taken into consideration by standard delineation. Despite the margins around the delineated areas, 42 of these 67 nodes (63%) would not have received at least 95% of the prescribed doses and would have been mistreated [84]. Figure 2 illustrates this point thanks to examples of atypical drainage in some prostate cancers. There is a need to change the paradigm of “one-size-fits-all”. SLNM is a path to explore, but some questions remain: would it be better to only treat draining nodes that are identified to reduce the irradiated volumes and what benefits would there be in terms of decreasing morbidity? Alternatively, would it be best to treat the nodes identified in addition to the usually described nodal areas and what would the oncological outcomes be? Clinical trials at a larger scale to answer these questions are lacking.

In conclusion, SLNB is a cornerstone for the management of penile cancers from a surgeon’s point of view, but its use in radiation oncology is almost inexistent, as the use of radiotherapy to treat lymph nodes has no validated indications at present. For bladder, testicular and renal cancers, the literature is much sparser, and the performances of SLN techniques in these indications remain very uncertain and experimental. SLNM and SLNB are still under investigation for prostate cancers, but the results are more promising and could help to better define irradiation volumes for more personalized radiotherapy.

## 5. Anal Cancer

### 5.1. SLNB Shows Better Performances Than FDG PET-CT for Detecting Metastases in Inguinal Nodes

The standard treatment of anal cancers is based on radiotherapy for T1 N0 tumors or concurrent radiochemotherapy (most often with 5FU and mitomycine C) for the others. Irradiation concerns the gross tumor, pelvic nodes, and inguinal nodes. Cancer staging currently relies on FDG PET-CT due to its high sensitivity. For instance, a study showed the perfect sensitivity of PET-CT, which did not miss any metastatic inguinal nodes, but reported a significant number of false-positive images, leading to a poor positive predictive value of only 43% [85]. Another study evaluated the SLNB of inguinal nodes and found this technique to be superior to FDG PET-CT, with fewer false-positive and false-negative patients [86]. In addition to better accuracy, a study revealed that pSLN was associated with oncological outcomes and a much better prognosis factor than positive inguinal uptake in FDG PET-CT. In fact, inguinal pSLN was significantly associated with a decrease in disease-free (21 vs. 56 months; *p* = 0.046) and overall (28 vs. 59 months; *p* = 0.028) survival [87]. Inguinal SLNB should be used more [88], as several literature reviews have reported good reproducibility and performance and acceptable rates of complications, but its deployment is limited by a lack of trials with a large population, notably because anal cancer is a rather rare cancer [89,90].

### 5.2. SLNB Could Spare Groin Irradiation and Its Toxicities

The selection of patients for groin irradiation currently depends on tumor size: T1 tumors are not a systematic indication, while groin irradiation is generally indicated for T2 or higher tumors. These rules present two problems: first, some T1 tumors may have occult inguinal metastasis whereas some T2 tumors may not, and second, groin irradiation can be poorly tolerated. The idea of adjusting radiation fields based on SLNB is not new in anal cancers [91]. A pilot study tested the feasibility of performing inguinal SLNB on patients with T1 or T2 anal tumors and irradiating the groin only in cases of pSLN [92]. The results of SLNB changed management in half (10/20) of their patients: 4 patients with a T1 tumor and pSLN received groin irradiation that was not initially indicated, and 6 patients with a T2 tumor and nSLN avoided groin irradiation that was initially indicated. Nevertheless, treatment de-escalation requires caution because a prospective study agreed with the feasibility of SLNB but also reported the cases of 2 out of 14 patients with nSLN who were spared groin irradiation, and who then developed inguinal metastasis at one year and two years, respectively [93]. Another study combined the use of FDG PET-CT and SLNB for inguinal staging, and patients presenting no sign of inguinal involvement in both exams avoided groin irradiation, and then presented significantly less inguinal dermatitis, especially severe dermatitis (grades 1–2: 12% vs. 50% and grades 3–4: 0% vs. 17%; *p* < 0.05) [94]. A retrospective study confirmed the difference between patients with nSLN and pSLN in terms of prognosis for disease-free and overall survival, and showed that it seemed safe not to target inguinal nodes in cases of nSLN, as none of their patients presented inguinal recurrence after a mean follow-up of 43 months [95].

In conclusion, disease staging in anal cancers is currently based on FDG PET-CT, which has shown good performances for pelvic nodes or visceral metastasis. However, FDG PET-CT has its limits for inguinal status, with a high rate of false positives. Inguinal SLNB can be seen as a more reliable alternative to inguinal staging, as well as to “LFGRT” where radiation fields are tailored to each patient. As anal cancers are uncommon, data on oncologic outcomes are lacking and comparative trials are needed.

## 6. Head and Neck Cancers

### 6.1. SLNB Is a Promising Procedure for Head and Neck Cancers

SLNB appears to be a well-accepted technique at present, providing a significantly lower surgical and postoperative morbidity when compared to elective neck dissection [96]. Several studies showed the performance of SLNB, leading to a meta-analysis of 26 studies (and 766 patients) by Thompson et al., who found a pooled sensitivity and a negative predictive value of 95% and 96%, respectively, for all head and neck cancers, and of 94% and 96%, respectively, for the subgroup of oral cavity cancer [97].

A recent randomized phase III trial further demonstrated an oncologic equivalence between SLNB and neck dissection in T1-T2 N0 oral and oropharyngeal cancers, with no significant difference at 2 and 5 years in neck node recurrence, disease-specific mortality, or overall survival [98]. A retrospective cohort study of 816 patients with a lateralized or paramedian early-stage oral cancer even suggested better outcomes with SLNB compared to ipsilateral elective neck dissection. The results showed more contralateral regional recurrences in patients undergoing ipsilateral neck dissection versus SLNB (3.8% vs. 1.3%; *p* = 0.018), and statistical analyses found a significantly higher risk of these patients presenting with contralateral recurrence (Hazard Ratio = 2.585; *p* = 0.030). The patients with contralateral node recurrence seemed to have a worse prognosis than those whose occult contralateral metastasis were detected earlier thanks to SLNB, with a disease-specific survival rate at five years of 42% versus 88% (*p* = 0.066) [99].

### 6.2. Nodal Irradiation Is a Key Treatment in Head and Neck Cancers

In contrast to surgeons, radiation therapists treat head and neck cancers that can be more advanced in terms of size (T1 to T4 tumors) and nodal involvement. Irradiation is delivered to both sides of the neck in most cases. This is a long-standing practice [100,101], mainly because empirical convention is used to avoid contralateral recurrence, rather than evidence-based medicine [102,103]. However, it is now well-documented that bilateral irradiation engenders more frequent and more severe acute and late toxicities, degrading the patients’ quality of life due to complications such as fibrosis, dysphonia, xerostomia, sticky saliva and dysphagia [104,105,106,107].

In order to avoid either over- or under-treating, the applications for SLNM and SLNB were explored in the field of radiotherapy with two aims: to determine tumor lymphatic drainage in each individual patient and to detect occult metastasis. Some prospective phase I-II monocentric studies tested LFGRT, which consisted of targeting only the lymph node levels with radiotracer fixation. De Veij Mestdagh et al. first evaluated SLNM in patients with lateralized cT1-3 N0-2b head and neck cancers thanks to radiolabelled 99mTc-colloid, and found that contralateral drainage affected 11/54 (20%) of the patients involving node levels II (88%), III (25%), and IV (13%), and was significantly associated with T3 tumors compared to T1 and T2 tumors (45% vs. 14%; *p* = 0.035) [108]. They then showed some dosimetric benefits to only treating the mapped drainage areas with median dose reductions in the contralateral parotid and submandibular glands, larynx, and thyroid gland [109], which translated into clinical benefits with significant reductions in dysphagia, gastric tube placement, and late xerostomia [110]. In the studies cited above, the patients did not undergo a pathological examination of their hot nodes, so they designed the ongoing SUSPECT-2 trial for patients with cT1-4 N0-2b lateralized tumors, where a pathological examination is performed in case of contralateral SLN, and bilateral neck irradiation will be indicated only if malignant involvement is histologically proven [111]. A Russian study of 26 patients with a cT1-2 N0 tongue cancer found 10/26 (38.5%) patients with bilateral drainage. The experimental LFGRT plans made significant reductions possible in both the irradiated volume and the mean doses received by the spinal cord, as well as the contralateral parotid gland, when compared to virtual plans for bilateral irradiation according to standard guidelines [112]. A Belgian study of 44 patients with cN0 head and neck cancers found 48% of patients with unilateral drainage and 16% with unexpected drainage that would not have been covered by usual delineations. The experimental plans also showed some dosimetric benefits and had positive clinical outcomes for dysphagia, xerostomia, hypothyroidism and patients’ quality of life [113].

In conclusion, SLN techniques are used increasingly in head and neck cancers and will become a major indication. This could be an opportunity for the use of “LFGRT” to decide between a unilateral or bilateral nodal irradiation, and to determine which drainage areas should be included or spared.

## 7. Discussion and Future Directions

The role of SLN in treatment planning presents a certain level of interest and has some feasibility, but we may wonder how to use it concretely in daily practice. An important step would be to conduct clinical trials comparing standard delineated volumes and tailored volumes. The objectives would be to decrease radiation toxicities and improve oncological outcomes by treating unexpected drainage areas, while noting that reducing the volumes too much could be a pitfall. A difficulty would be calculating each indication to determine which would be most effective and detect a difference in efficacy and/or toxicity. Table 1 indicates the main ongoing trials that will provide some answers.

This review aimed to provide an overview of the use of SLN techniques to individualize radiotherapy treatment by reviewing the current knowledge, but is limited in its power to reach definitive conclusions, given the sparse literature. The impact of SLNM is difficult to accurately evaluate and will differ according to the localizations. It should be most significant in head and neck cancers, since many studies showed both dosimetric and clinical benefits with decreased toxicity and could be routinely used. It should also be meaningful to avoid loco-regional recurrence in prostate cancer, with several studies showing better target coverage. In other localizations, we acknowledge that its impact is more theoretical, because the majority of studies found in the literature only show dosimetric improvements, while studies verifying whether such results will translate into clinical outcomes are lacking.

Another point that should be stressed is the need for a strong collaboration between the different actors and departments (notably surgery, nuclear medicine and radiotherapy) to obtain a smooth workflow. Additionally, developing studies on lymph-flow-guided radiotherapy is not a simple task, since surgical studies are needed in order to first demonstrate that the indications of SLN are safe and effective. This could be a great opportunity to reinforce collaboration between the oncological specialties and to conduct more joint studies.

## 8. Conclusions

After providing a major breakthrough for surgeons, SLN techniques have shown promising results as far as radiotherapy treatment planning is concerned. SLNM and SLNB appear to be safe procedures that help redefine the irradiation volumes that should be used for each individual, avoiding the use of a probabilistic method, thus avoiding under- or over-treatment. Some indications have a head start, such as head and neck cancers, where several trials can be found in the literature, whereas other indications are still theoretical, as only retrospective or in silico studies have been published. In any case, “LFGRT” cannot be used in routine treatments for individualization of the treatments, and should currently be seen as an exploratory technique, given the lack of published phase III randomized trials or meta-analyses with large numbers of patients. It nonetheless remains an inspiring concept, which should be further developed soon.

## Figures and Tables

**Figure 1 cancers-15-02736-f001:**
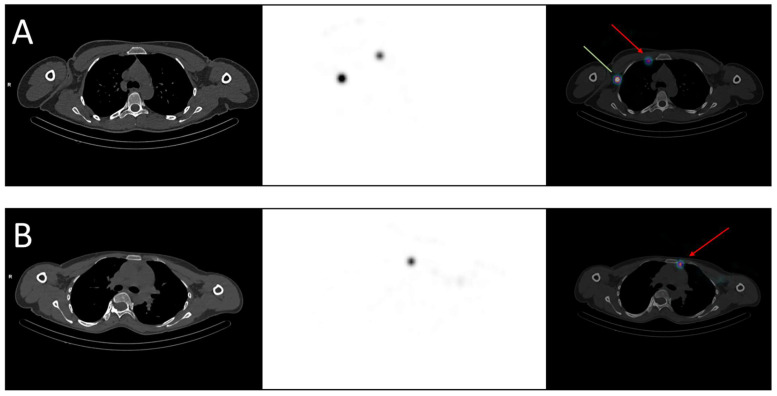
Examples of breast cancer drainage in the internal mammary nodes visualized in SPECT-CT. (**A**) A 36-year-old woman was diagnosed with two malignant nodules in the right breast, localized in the inner quadrants. The sentinel lymph node mapping revealed drainage in both the axillary (green line) and the internal mammary nodes (red arrow). (**B**) A 40-year-old woman was diagnosed with a malignant nodule in the left breast, localized in the upper-inner quadrant. The sentinel lymph node mapping revealed drainage in the internal mammary nodes (red arrow).

**Figure 2 cancers-15-02736-f002:**
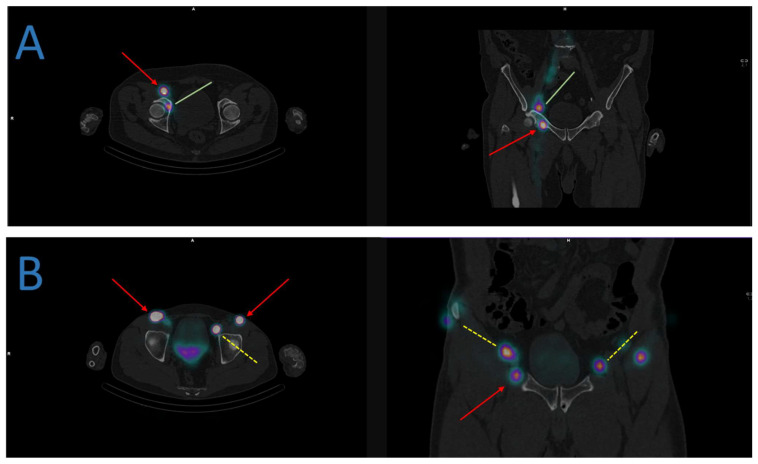
Examples of atypical lymphatic drainage in some prostate cancers. (**A**) The sentinel lymph node mapping identifies a drainage in the right obturator (green lines) and inguinal nodes (red arrows). (**B**) The sentinel lymph node mapping identifies a bilateral drainage in the external iliac (yellow dashed lines) and inguinal nodes (red arrows).

**Table 1 cancers-15-02736-t001:** Ongoing trials evaluating the role of the sentinel lymph node in treatment planning in radiotherapy. We searched for currently active trials registered on ClinicalTrials.gov with the following tags: “sentinel” and “radiotherapy”. Abbreviations: head and neck squamous cell cancer (HNSCC).

ClinicalTrials.gov Identifier	Name	Cancer Localization	Phase	Estimated Population	Start Date	Expected Completion
NCT02642471	PHENIX/CSEM010	Cervix	III	1080	December 2015	December 2022
NCT03968679	SUSPECT-2	HNSCC	II	90	July 2019	August 2025
NCT04577950	SENNAN	Endometrial	II	120	January 2021	December 2024
NCT05076942	GROINS-V III	Vulvar	II	157	January 2021	January 2029
NCT04688528	SEMIRAHN	HNSCC	II	147	June 2021	January 2027
NCT04594187	MelPORT	Melanoma	II	168	August 2021	February 2025
NCT04073706	ENDO-3	Endometrial	III	760	January 2022	January 2031
NCT05040815	INSPIRE	Anal	II	45	June 2022	June 2028
NCT05333523	PRIMO	HNSCC	III	242	October 2023	October 2028

## Data Availability

No datasets were created or analyzed to write the present article.

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
