# Peer review of "The Sentinel Lymph Node in Treatment Planning: A Narrative Review of Lymph-Flow-Guided Radiotherapy"

_cancers, 2023, doi:10.3390/cancers15102736_

Round 1

Reviewer 1 Report

The authors present a comprehensive review of the role of sentinel nod biopsy (SLNB) in the treatment of breast cancer, gynecologic and urologic tumors, anal cancer and head and neck tumors. The review is interesting and based on recent studies. The main task of the article is an evaluation of sentinel node mapping and sentinel lymph node biopsy for navigation of tailored radiotherapy. The influence of results of SLNB on indication of radiotherapy and the influence of sentinel lymph node mapping on target volume are discussed for each type of the tumor and the discussion is supported by both already published and ongoing studies, altkough ther number is limited. 

Because the title is "The sentinel lymph node in the treatment planning: a narrative review of lymph flow guided radiotherapy" , more details about the impact of sentinel lymph node mapping on the radiotherapy technique could be useful including ilustrative radiotherapy plans (at least for breast cancer and prostate cancer). 

The article is well writen and the topic is modern. After minor revisions mentioned above I suggest to accept. 

Author Response

Estimated reviewer,

We are grateful for your comments.

Please see the attachement for our responses.

We thank you.

Best regards.

Reviewer 2 Report

In my opinion, the analyzed topic is interesting enough to attract the readers’ attention. The aim of this review is to highlight the feasibility and level of proof to use the sentinel lymph node technique for treatment planning in radiation oncology. I think that the abstract of this article should be improved highliting the aim of the study.

In my opinion, the discussion could be studied in depth and extended. Maybe, it could be useful the evaluation of the state of the art of  the role of lymphadenectomy in gynecological cancers. Also could be interesting, the evaluation of possible new treatment strategies thanks to the detection of micrometastasis in lymph nodes. In particular I suggest these two articles  PMID: 32036457 and PMID: 35598492, respectively, to get deeper in the topic.  Moreover, the structure and layout of the article should be revised. In particular, the reference part. Because of these reasons, the article should be revised and completed. Figures and tables are clear. Considered all these points, I think it could be of interest for the readers and, in my opinion, it deserves the priority to be published after minor revisions.

A moderate review of English language and grammar should be performed

Author Response

(The authors gave the same response as above.)

Reviewer 3 Report

In the presented article, the authors, based on selected literature items analyzed the possibility of using SLNM and SLNB in a more personalized radiotherapy in terms of identifying atypical lymph drainage to the sentinel node and the possibility of reducing irradiated nodal volumes in tumors of various locations. The article is a summary of the current state of knowledge on this subject.

I am posting a commentary below, with two small notes regarding the literature and the section on vulvar cancer.

Data on the anatomical variants of lymph drainage to the sentinel node from studies on cadavers or surgical, intraoperative assessment, both previously and currently published are scarce (e.g. Clinical & Experimental Metastasis 2022; 39:139–157, CANCER 2002; 95, 2: 354-360), nevertheless, it may be worth mentioning this, and also that information on anatomical variants is now more available (theoretically for any patient with a non advanced cancer, given a sufficiently widespread SLNM). And the benefits of this seem obvious.                                                                                                           The topic of broadly understood personalization is very current and desirable in modern oncology. The use of SLNM to more precisely determine the irradiated volumes is the subject of a number of works. In clinical practice, contouring of nodal areas is based primarily on available atlases.

The analysis of the literature conducted by the authors shows that SLNM may be helpful in the selection of breast cancer patients requiring irradiation of internal mammary nodes (which is associated with an increased risk of cardiological and pulmonary toxicity). In turn, the ARM may help to save the axillary nodes to which the lymph from the upper limb flows.

In the section on vulvar cancer, the authors recall the most important clinical studies that have established the role of SLNB in the diagnosis of lymph nodes. In the part regarding the management of ipsilateral groin, I would add for the sake of order that this applies not only to FIGO IB to II stages, but also to lateral lesion (≥ 2 cm from vulvar midline).

In the section on head and neck cancers, the authors cite the literature showing the growing role of SLN techniques, which in selected cases may allow the volume of irradiated nodes to be limited to one side. Few centers conducting radiotherapy use SLNM to consider limiting the volume of irradiated nodal fields in clinical practice. From oral information, from friendly centers, I know that this applies to, for example, lateral localization in non-advanced buccal mucosa cancer or tonsil cancer with favorable histology.

Author Response

(The authors gave the same response as above.)

Reviewer 4 Report

Title of the work. The sentinel lymph node in treatment planning: a narrative review of lymph flow guided radiotherapy. It is a well-written, interesting, and important material, dealing with the role of sentinel node biopsy and the potential effect of the result of the sentinel node procedure (sentinel node mapping) on radiotherapy treatment planning. Nevertheless, I did not find any novelty or any special message in it. Over the undoubted values of this summary, it is a kind of a mixture of surgical and/or radiotherapy consideration and a mixture of „radiotherapy relevant” and non-relevant cancer sites/organs, with omission of some important indications of sentinel node biopsy (see melanoma of the skin, Merkel cell carcinoma etc.). The authors list some well-known indication/knowledges and of course the material describes some interesting future project about lymph node mapping-based radiotherapy (e.g., axillary reverse node mapping), but most of them theoretical and the relevant clinical studies are ongoing. (The decreased morbidity after limited field radiotherapy is an evident finding, however the relapse/survival effect is the final measure of value.) Conclusively, over the real values of the material, in this form I do not recommend for publication. I suggest a kind of a paper reconstruction aiming only the radiotherapy aspects and the relevant cancer diseases and discussing the present knowledge/experiences about “need to change the paradigm of “one-size-fits-all”.

Author Response

(The authors gave the same response as above.)

Round 2

Reviewer 4 Report

I absolutely accept the answers/comments of the authors. I consider the manuscript a well-written and important summary, however, till date I miss the present clinical consequence/novelty in it. Nevertheless, I appreciate the corrections and explanations of the authors, and the addition of thoughts about the uncertainties/future directions, moreover, to tell the truth this work really the first review in this topic.

To return the cancer categories, I suggest an introduction sentence/explanation about the examined categories, since e.g., in renal cancers RT has a limited role as well, like in melanoma malignum.

Author Response

Dear reviewer,

Many thanks for your fast answer.

Point 1: I absolutely accept the answers/comments of the authors. I consider the manuscript a well-written and important summary, however, till date I miss the present clinical consequence/novelty in it. Nevertheless, I appreciate the corrections and explanations of the authors, and the addition of thoughts about the uncertainties/future directions, moreover, to tell the truth this work really the first review in this topic.

Response 1: We agree that this review wil not be sufficient to routinely perform "lymph flow guided RT", since no new data or prospective trial is presented. We tried nonetheless to encourage its development.

Point 2:To return the cancer categories, I suggest an introduction sentence/explanation about the examined categories, since e.g., in renal cancers RT has a limited role as well, like in melanoma malignum.

Response 2:We added in the introduction: "We will discuss localizations of cancers for which the SLN has been evaluated in surgery, and in which radiotherapy could play a role, being well-established like in head and neck cancers, or under investigation like in renal cancers." In fact, renal cancer was historically considered as radioresistant. However, thanks to dose escalation (with stereotactic irradiation for instance), renal cancers could be irradiated. We currently treat T1a tumors since it was validated, and we could also treat T1b tumors. Furthermore, future studies aim to evaluate RT for larger tumors as an alternative to surgery, or as an adjuvant treatment, and nodal irradiation will be rediscussed. So, we decided to cite RT in renal cancers that has a growing role and several ongoing studies.

Best regards